# Quantifying Coordination and Variability in the Lower Extremities after Anterior Cruciate Ligament Reconstruction

**DOI:** 10.3390/s21020652

**Published:** 2021-01-19

**Authors:** Sangheon Park, Sukhoon Yoon

**Affiliations:** Motion Innovation Center, Korea National Sport University, Seoul 05541, Korea; ptl2503@knsu.ac.kr

**Keywords:** coordination, variability, continuous relative phase, anterior cruciate ligament reconstruction

## Abstract

Patients experience various biomechanical changes following reconstruction for anterior cruciate ligament (ACL) injury. However, previous studies have focused on lower extremity joints as a single joint rather than simultaneous lower extremity movements. Therefore, this study aimed to determine the movement changes in the lower limb coordination patterns according to movement type following ACL reconstruction. Twenty-one post ACL reconstruction patients (AG) and an equal number of healthy adults (CG) participated in this study. They were asked to perform walking, running, and cutting maneuvers. The continuous relative phase and variability were calculated to examine the coordination pattern. During running and cutting at 30 and 60°, the AG demonstrated a lower in-phase hip–knee coordination pattern in the sagittal plane. The AG demonstrated low hip–knee variability in the sagittal plane during cutting at 60°. The low in-phase coordination pattern can burden the knee by generating unnatural movements following muscle contraction in the opposite direction. Based on the results, it would be useful to identify the problem and provide the fundamental evidence for the optimal timing of return-to-sport after ACL reconstruction (ACLR) rehabilitation, if the coordination variable is measured with various sensors promptly in the sports field to evaluate the coordination of human movement.

## 1. Introduction

Anterior cruciate ligament (ACL) injuries are the most frequent of sporting injuries, occurring during dynamic movements such as a cutting maneuver. They are generally caused by excessive internal rotation of the tibia against the femur, resulting in increased stress and tension on the ACL [1,2,3]. Such injuries result in a variety of knee joint problems and excessive medical and rehabilitation costs [4,5]. In the United States, ACL injuries are common, with 250,000 incidents occurring each year and more than 130,000 ACL reconstruction (ACLR) cases [6]. ACLR refers to surgery that secures the graft in the same position as the existing ACL to generate ligament function; it is the most commonly performed surgery in ACL injuries [7,8]. Essentially, ACLR is aimed to restore joint stability and the original level of physical activity. However, only 65% of patients recover to their pre-existing daily living standards, and 50% of them can return to their prior sporting level [9,10]. Unfortunately, even for the patients who regain their pre-injury level of performance, there is a high risk of reinjury [11].

Functional disorders following ACLR have been cited as the reason for failure to return to pre-injury levels of daily life or sports [9]. Typical functional disorders experienced after ACLR include muscle atrophy, change in muscle elasticity around knee joint, reduced static and dynamic stabilities, defined as the ability of a body to return to equilibrium after being displaced, and reduced coordination, which is combination of body movements created with the biomechanical parameters, due to a structural change in the knee joint. That is, despite sufficient recovery of muscle strength through surgical treatment and rehabilitation, various function is not possible, because the knee may not be fully functional when performing movements after ACLR. According to studies, although more than 90% of muscle strength (e.g., quadricep and hamstring muscle) was recovered through rehabilitation after reconstruction, if the original movement could not be performed, the cause was attributed to the lack of muscle activation due to the absence of a neuromuscular response. It has been reported that the more difficult the movement, the more severe the dysfunction could be [1,12,13,14,15,16,17,18]. Therefore, it is important to understand the lower extremity joints in order to prevent recurrent ACL injuries and to help patients return to daily life.

Despite the successful completion of ACLR and rehabilitation, constant dysfunction hinders locomotion required for a healthy living [4,19,20,21]. According to a study that examined human locomotion-related biomechanical changes, the ACLR group demonstrated greater knee flexion angles and moments in the sagittal plane during walking than those of the healthy group [22,23]. Conversely, it has also been suggested that ACLR patients exhibit smaller knee flexion angles and moments, causing interpretation difficulties [24,25,26]. Furthermore, as a result of examining biomechanical variables on the frontal plane when walking, the ACLR group reported a smaller varus angle than the healthy group, with no difference in moment [27]. In a study measuring biomechanical variables on the transverse plane, the ACLR group demonstrated a smaller internal rotation angle than that of the healthy group during the stance phase and a larger internal rotation angle during the swing phase, with a small external rotation moment [27,28,29]. Additionally, a study that examined biomechanical changes in running found a larger varus angle during the stance phase and an excessive external rotation angle of the knee in the respective ACLR group compared with that of the healthy group [30,31]. Lastly, a study on cutting maneuver reported that the ACLR group showed a smaller knee flexion and greater valgus angles and moment than that of the healthy group [32,33]. The results of previous studies are inconsistent to identify the characteristics of the ACLR population. It makes it difficult to define what characteristics the ACLR group has and their problems after surgery. In addition, biomechanical differences in previous studies that occur after ACLR are indicative of functional changes in movement, and the amount of change varies depending on the movement intensity. However, studies have only considered variables in single parameter or a single joint of the movement such as the single joint angle or moment to study the characteristics of dysfunction occurring after reconstruction, while disregarding the movement difficulty and joints’ interaction. Therefore, to examine overall movement coordination characteristics after ACLR according to variations in movement difficulty, it is necessary to identify both coordination patterns between joints at each movement difficulty level and the degree of variability to indicate consistency and repeatability [34,35,36].

Different methods for quantifying coordination patterns and variability have been suggested in many studies [34,35,37,38]. The concept of coordination was first shown for bimanual rhythmic coordination by Kelso [36]. When performed to rhythmically move both left and right index fingers and hands, it was found that patients could only produce consistent and repetitive coordination patterns at relative phase (in-phase or out-of-phase). Eventually, two joints (or segments) in coordination can be modelled by two oscillators that are non-linearly coupled. The continuous relative phase (CRP) is a method of expressing angular displacement and angular velocity produced from two different joints in the phase angle. CRP explains the interaction between each joint using the relative difference between the phase angles, and its standard deviation is used to express the variability of the coordination pattern. Additionally, as it reflects the entire phase rather than a specific event in interpreting motion, it can address the limitations of previous studies that were unable to fully capture the characteristics of spatiotemporal movement [39,40].

In order to record kinematic parameters of a human movement, it is essential to use a feasible and reliable measurement system. In motion analysis, the use of sensors as well as an infrared camera is widespread. Specifically, sensors are easy to use, portable, and cost efficient. However, most sensors are prone to failure to capture precise motions in three dimensions. In contrast to that, an infrared camera cannot be moved and is expensive during an experiment, but this system is well known for high reliability in three dimensions. Hence, prior to using a sensor-based system in motion analysis, it is necessary to use an infrared camera, which is a highly reliable tool to determine the differences in coordination patterns in three dimensions.

Therefore, the purpose of this study was to quantitatively determine the changes in lower extremities coordination patterns (CRP and CRP variability) according to ACLR experience and movement difficulty. This study will be conducted in three hypotheses, including (1) coordination patterns show clear and consistent result, (2) coordination patterns are different at each movement, and (3) coordination patterns are different at each movement plane.

## 2. Materials and Methods

### 2.1. Participants

Twenty-one post ACLR male participants (AG) and the same number of healthy males (CG), who had no history of lower limb injury and pain, participated in this study (Table 1). No participants report experience of a slight injury within 6 months. The inclusion criteria for all of AG included undergone ACLR at least 12 months prior and obtained a score of 80 or higher on the knee injury and osteoarthritis outcome score (KOOS), indicating complete medical recovery [41,42]. Required sample size was estimated base on the previous kinematic study conducted on ACLR patients [43,44] with the minimum statistical power being 80% (effect size = 0.80). G-Power software (version 3.1.9.2, Kiel University, Germany) was used in the power and sample size calculation. The Korea National Sport University Institutional Review Board approved all experimental procedures (20180625-017), and written informed consent was obtained from all individuals participating in this study.

### 2.2. Procedure and Data Collection

After a full warm-up, participants were asked to perform walking, running, and cutting maneuvers. Walking and running were performed with preferred speed, and cutting maneuvers were performed at angles of 30° and 60° with their maximum speed (preferred speed; walking, CG: 1.42 ± 0.09, AG: 1.45 ± 0.12; running, CG: 3.39 ± 0.18, AG: 3.30 ± 0.11; *p* > 0.05). In cutting maneuver trials, participants used direct cutting, in which the subject does not cross the legs but extends straight to change direction following the line tape on the floor. Participants were instructed to perform cutting maneuvers along the line of tape placed five meters away from starting line. For the pivot feet, AG and CG used ACLR side feet and dominant feet, respectively. Ten successful trials were collected each trial condition and were used in data analysis. Sufficient rest time was provided between each trial, and all conditions were balanced and randomized in order.

For 3D motion analysis, a total of 16 retro-reflective markers and two clusters were affixed to the body. Markers included right/left iliac crest, right/left anterior superior iliac spine, right/left posterior superior iliac spine, right/left greater trochanter, thigh 1–4 (cluster), medial/lateral epicondyle, shank 1–4 (cluster), medial/lateral malleolus, meta 1 (toe), navicular, mid-foot, heel (Figure 1).

All trials of each condition were recorded with eight infrared cameras (Oqus 300+, Qualiysis, Göteborg, Sweden; 200 Hz; resolution, 1280 × 1024 pixels). The cameras were positioned around a moving space and were calibrated using the nonlinear transformation (NLT) method (Figure 2). The overall camera reconstruction error was 0.15 mm for camera calibration area. A static trial was recorded to provide a baseline for the dynamic trials. All medial markers were removed prior to the collection of the dynamic trials.

### 2.3. Data Processing and Variable Calculation

The marker coordinates were obtained through automatic marker tracking using Qualisys Track Manager (Qualisys, Sweden). Three-dimensional marker coordinates were reconstructed based on the digitized marker coordinates. The reconstructed marker coordinates were subjected to filtering to reduce the random experimental errors. A zero-phase lag 4th-order Butterworth low-pass filter was used, and the cutoff frequency at that time was set at 99% by analyzing the power spectrum density [45].

Lower extremity joint centers were computed based on the location of surface markers. The mid-points between the medial/lateral malleolus and epicondyles were used as the ankle and knee joint centers, respectively. The hip joint center was computed by the methods proposed by Tylkowski [46]. A 3-segment (foot, shank, and thigh) lower rigid-body model was used in this study. Segments were defined by appropriate proximal and distal joint center except the foot, which was defined by toe and heel markers. Additionally, the segment vectors and segment reference frames were defined. The relative orientation angles of the segments to their respective linked proximal segments were computed from the orientation matrices from the global to the segmental reference. The XY’Z’’ rotation sequence was assumed in the decomposition of the matrices for the computation of the orientation angles. Each lower extremity joint angle was set at positive for flexion and negative for extension with respect to the *x*-axis, positive for abduction and negative for adduction with respect to the *y*-axis, and positive for external rotation and negative for internal rotation with respect to the *z*-axis (Figure 1). Angular velocities of the orientation angles were computed based on the first derivatives of the angles in Visual 3D software (C-Motion, Inc., Germantown, MD, USA).

After calculating the angular displacement [*θ*] and angular velocity [*ω*] of each joint along each plane at each data point *i*, the *θ* and *ω* of each analyzing phase were interpolated to 100%. Then, the horizontal and vertical axes were calculated using Formulas (1) and (2) to standardize them to +1 and −1 based on the maximum and minimum values.
(1)$$\mathrm{Horizontal}-\mathrm{axis}:\;\theta_{i}=\frac{2\times \left[\theta_{i}-\min\left(\theta_{i}\right)\right]}{\max\left(\theta_{i}\right)-\min\left(\theta_{i}\right)}-1$$
(2)$$\mathrm{Vertical}-\mathrm{axis}:\;\omega_{i}=\frac{\omega_{i}}{\max\left(\left|\omega_{i}\right|\right)}$$

The normalized angular displacement and angular velocity were converted from the cartesian coordinate system to the polar coordinate system, and the phase angle of each joint within the polar coordinates was calculated to be within 0–180° using Formula (3) with respect to the right horizontal axis. The calculated phase angle of each joint was expressed as CRP based on the difference between the two values, as shown in Formula (4) [39].
(3)$$\mathrm{Phase}\;∠:\varphi =tan^{-1}\left(\frac{\omega_{i}}{\theta_{i}}\right)$$
(4)$$CRP_{i}=\left|\varphi_{proximal}-\varphi_{distal}\right|$$

The average of the absolute CRP value was expressed within the range of 0–180°; value closer to 0° indicates a similar coordination pattern (in-phase), whereas closer to 180° indicates a different coordination pattern (out-of-phase). Additionally, the standard deviation of CRP was calculated to compute the quantitative values for the variability of the coordination pattern. Herein, the standard deviation of CRP was a value representing the variability of the entire period of each movement performed by the subject, and the standard deviation of each point on the ensemble curve of CRP was calculated and expressed as variability in MATLAB software (The MathWorks, Inc., Natick, MA, USA) (Figure 3) [47].

### 2.4. Statistical Analysis

Independent t-test was performed to verify the difference in lower limb coordination pattern (CRP and CRP variability) during the stance phase of walking, running, and cutting (30° and 60°) between the two groups with regard to ACLR. The significance level was set at *α* = 0.05.

## 3. Results

This section may be divided by subheadings. It should provide a concise and precise description of the experimental results, their interpretation, as well as the experimental conclusions that can be drawn.

### 3.1. CRP in Hip–Knee and Knee–Ankle Joints by Difficulty of Exercise

The difference between the two groups in the knee–hip CRP during walking (Table 2) was nonsignificant. During running, the hip–knee CRP was significantly different between the groups in the sagittal plane, while the knee–ankle CRP was significantly different between the groups in the transverse plane (Table 2; *p* < 0.05). Furthermore, during cutting at 30°, the hip–knee CRP was significantly different between the groups in the sagittal plane (Table 2; *p* < 0.05). During cutting at 60°, the hip–knee CRP was significantly different between the groups in the sagittal plane (Table 2; *p* < 0.05), and the knee–ankle CRP revealed a significant difference between the groups in the frontal plane (Table 2; *p* < 0.05) (Figure 4 and Figure 5).

### 3.2. CRP Variability in Hip–Knee and Knee–Ankle Joints by Difficulty of Exercise

The CRP variability in the hip–knee joint was not significantly different between the groups during walking, running, or 30° cutting. However, a significant difference in the CRP variability in the hip–knee joint couple between the groups during 60° cutting in the sagittal plane was noted (Table 3; *p* < 0.05). However, the CRP variability in the knee–ankle joint was not significantly different between the groups (Figure 4 and Figure 5).

## 4. Discussion

In this study, the difference in lower limb coordination patterns were investigated based on movement difficulty following ACLR using CRP and CRP variability. The results of this study found no difference in the hip–knee and knee–ankle joint CRP values during walking (Table 2 and Table 3, *p* > 0.05). This may suggest that it is difficult to determine differences from the original movement when performing easy and daily routine movements such as walking compared to running and cutting. This finding is consistent with those of previous studies that revealed that differences in movement due to dysfunction may become more apparent as the difficulty of movement is increased [12,13,14,15,16,17,18,48].

In running, it was found that the hip–knee joint CRP of the AG was 18% greater than that of the CG in the sagittal plane, and the knee–ankle CRP of AG was 15% greater than that of the CG in the transverse plane (Table 2, *p* < 0.05). These results indicate that the hip and knee joints of the AG have a worse in-phase coordination pattern than those of the CG during running in the sagittal plane. As such, it means that the muscles are unable to contract in the corresponding direction in a synchronized manner and exhibit unstable coordination patterns [36,37,38]. The results also suggest that the knee–ankle joint coordination pattern in the transverse plane is closer to the out-of-phase coordination pattern in the AG and in-phase coordination in CG during running. The out-of-phase knee–ankle joint coordination pattern shown in AG implies that the muscles responsible for the transverse movement of the two joints are contracting in opposing directions [37,49,50] and can be considered a risk factor for articular system injury by inducing contradictory movements of the proximal and distal segments during the stance phase [1,2,3]. This is because of the torsion force on the knee joint caused by twisting between proximal and distal segments. This type of coordination pattern is a potential cause of re-injury following ACLR by causing excessive rotation of the femur with respect to the tibia during the stance phase [30].

We also found significant differences for cutting maneuvers between groups. The hip–knee joint CRP in the sagittal plane was greater during cutting maneuver at 30° and 60° in the AG compared with that of the CG while the knee–ankle joint CRP of AG was lower during cutting maneuver at 60° in the frontal plane (Table 2, *p* < 0.05). These results may suggest that the hip–knee joint coordination pattern of the AG in the sagittal plane has a worse in-phase coordination pattern during movements of greater difficulty than walking and running, such as 30 and 60° cuttings. As mentioned above, this indicates that the muscles do not contract in the corresponding direction and depict an unstable in-phase coordination pattern [37,49,50]. The hip–knee joint coordination pattern in the sagittal plane during cutting is responsible for shock absorption during the stance phase [51]. It generates an extension moment through simultaneous contraction upon initial contact, produces negative joint power through flexion, and later produces positive joint power and propels the body by continuously maintaining the extension moment [51]. Therefore, the out-of-phase coordination pattern of hip and knee joints showed in AG in the sagittal plane suggested that AG was not properly performed the coordination pattern for shock absorption, which is performed in CG. The improper coordination pattern of the hip and knee joints indicated that the hip joint fails to absorb shock. Instead, propulsion occurs during the initial contact phase, which results in an increased burden on the knee joint or the inability of knee extensors to activate sufficiently, causing sudden flexor movements without being able to withstand the impact completely, thus displaying an excessive out-of-phase coordination pattern during the initial contact phase. These results show that this coordination pattern can act as a precondition to ACL injury by failing to reduce the impact on the lower extremities and increasing the load on the musculoskeletal system through improper implementation of biomechanical action strategies that prevent adequate shock absorption from the ground [52].

In this study, the knee–ankle joint coordination pattern of the AG in the frontal plane during cutting maneuver at 60° revealed a greater in-phase coordination pattern than that of the CG; this was not observed during low-difficulty movements. Generally, humans have difficulty controlling certain movements as the degree of difficulty increases, which increases the probability of revealing inherent, fundamental problems [53]. When performing a dynamic 60° cutting maneuver, the dynamic valgus increases due to the influence of the internal direction of force applied to the center of mass. To reduce this force, the ankle inversion is required when changing directions [54]. However, the high in-phase knee–ankle coordination pattern of the AG observed in this study suggests that these joints perform movements in the same direction (knee valgus and ankle inversion) increasing the ACL’s tension and injury risk [1,2,3].

In addition, the hip–knee CRP variability of AG was lower than that of the CG during the cutting maneuver at 60° in the sagittal plane (Table 3, *p* < 0.05). Various studies have reported that variability will have a direct correlation with injury and that coordination patterns with low variability can be interpreted as performing a consistent movement cycle [33,34,37,55]. Yet, this can be interpreted pathologically as using only one action strategy and inflexible situational abilities due to fear, pain, or movement discomfort [34,35,37,38]. Therefore, the fact that the AG experiences fear, discomfort, and pain during difficult movements compared with the CG cannot be ruled out and can be inferred as the action strategy being limited due to this. In addition, the limited coordination pattern can be interpreted as a consistent strategy to minimize pain according to the situation, so that AG can be viewed as a positive coordination pattern to them, but this repetition of the coordination pattern exerts consistent load on a narrow-localized area resulting in damage to soft tissues, such as cartilage and ligaments, which can lead to negative long-term effects [35,49].

To date, a variety of studies have been conducted to verify the biomechanical differences resulting from ACLR, but conflicting results have made concluding difficult. In this study, it was judged that the problem was limited to a single joint, and the interpretation was continued. As a solution, a continuous relative phase (CRP) that can confirm the movement pattern of both joints was selected. The results of this study suggest that despite the structural regeneration and restoration of muscle strength after ACLR, these patients tend to resume daily life with an incomplete or compensated coordination pattern of the hip joint, which connects proximally to the knee joint. As such, the lack of an in-phase coordination pattern and the presence of an out-of-phase coordination pattern in AG can lead to ineffective absorption of impact absorption on the body and torsion of the joint. Additionally, as the difficulty of movement increases, a single coordination pattern is repeatedly used to increase the risk of soft tissue injury. Thus, although ACLR patients are considered to have reached a full clinical recovery, it is highly likely that the coordination pattern of the hip and ankle joints is not properly performed during movement, especially with increasing movement difficulty. To achieve a full functional recovery with an aim of resuming sports in participants with ACLR, it is necessary to minimize the negative ripple effects caused by different coordination patterns and improve the compensatory action that occurs when performing difficult movements. Furthermore, to rehabilitate the appropriate pattern, coordination training to minimize the load during more difficult movements is emphasized and recommended [56].

A limitation of the present study is that the CRP used in this study is a useful variable for evaluating the entire period of movement and identifying the characteristics of locomotion such as walking, running, and cutting maneuvers. However, it was confirmed that coordination patterns vary depending on the time point, even within the entire cycle (cycle to cycle, 0–100%). Therefore, if the phase can be dissected in more detail by applying and utilizing the advantages of CRP, which allows us to comprehensively observe each point in the entire movement, a more accurate point or circumstance at which the corresponding coordination pattern occurs can be determined. In future studies with CRP, detailed division of the phase to confirm the coordination pattern of motion according to the CRP cycle is suggested. Additionally, kinetic variable measurements (e.g., muscle activation, ground reaction force [GRF], joint moment) to examine the systematic movement and to investigate the relationship with coordination patterns would make for a more helpful study. The infrared camera used in this study is difficult to use and expensive, whereas the sensors such as the IMU (inertial measurement units) sensor, markerless motion solution, and video-based motion capture system are highly accessible and cost effective. Yet, these sensors do not have the reliability of three-dimensional data as high as the motion capture system with an infrared camera except for the IMU sensor. On two-dimensional data on the sagittal plane, the sensors usually show a sufficient reliability similar to the infrared camera. Therefore, to determine the rehabilitation progress or the timing of return-to-sport, the sensors can also be used on two-dimensional data on the sagittal plane on the field, as this study showed that there were distinctive differences in two-dimensional data on the sagittal plane. Such differences shown in the results were easily detected during running rather than walking, thus in future study, the reliability of the sensors should be evaluated in the sagittal plane during running. Additionally, the possibility of using the sensor was suggested according to the result without simultaneously examining the reliability of the sensor. It is necessary to measure infrared cameras and human movement analysis sensors, such as IMU sensors, at the same time to verify the use of the sensors on the field.

## 5. Conclusions

Our findings indicate that during increased difficulty, the in-phase and out-of-phase coordination patterns of the hip–knee and knee–ankle joints are lower in the AG than in the CG in the sagittal plane. Furthermore, the in-phase coordination pattern of the knee–ankle joint of the AG is higher than that of CG in the frontal plane, and the coordination pattern of hip–knee joint of the AG in the sagittal plane was limited. The heterogenous out-of-phase coordination pattern identified in the study can assume that there is a strain on the knee due to continuous unnatural and inappropriate movement patterns, which results in muscle contractions in the opposing directions. The limited hip–knee coordination pattern in the sagittal plane can increase repetitive loads by restricting the flexion/extension actions of the hip and knee joints, which play an important role in controlling the impact during the stance phase. The results shown in two-dimensional data on the sagittal plane in this study suggest the possibility to apply motion sensors as well as an infrared camera. The coordination and variability in lower extremities found in this study can be the key to determining the rehabilitation progress and the timing of return to sport on the field.

## Figures and Tables

**Figure 1 sensors-21-00652-f001:**
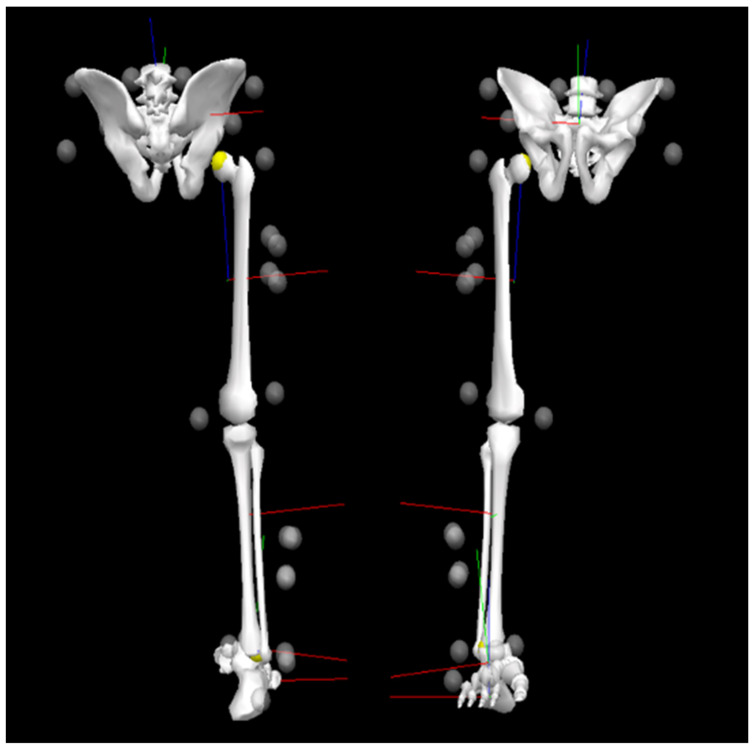
Attachment of markers on lower extremity (the axis of angular displacement and velocity were defined as Figure 2, red line = positive *x*-axis vector, green line = positive *y*-axis vector, blue line = positive *z*-axis vector, white circle = attachment of marker).

**Figure 2 sensors-21-00652-f002:**
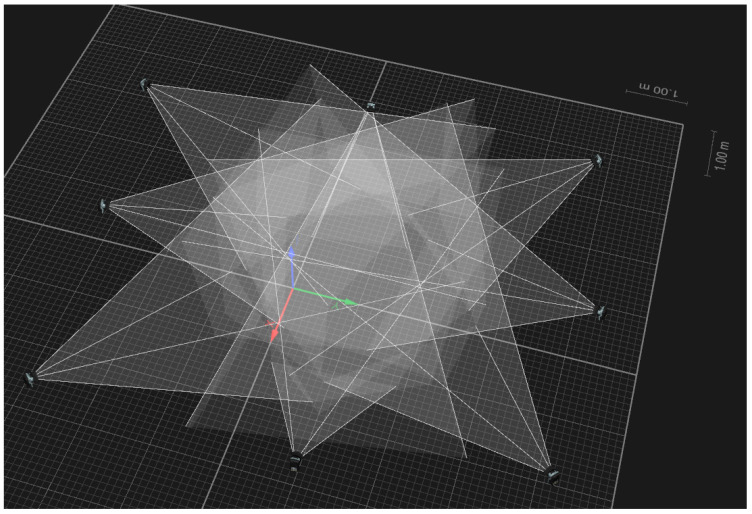
Three-dimensional capture area by Qualisys track manager (global coordination system, red line = *x*-axis, green line = *y*-axis, blue line = *z*-axis, shaded area = camera field-of-view of 3D cones).

**Figure 3 sensors-21-00652-f003:**
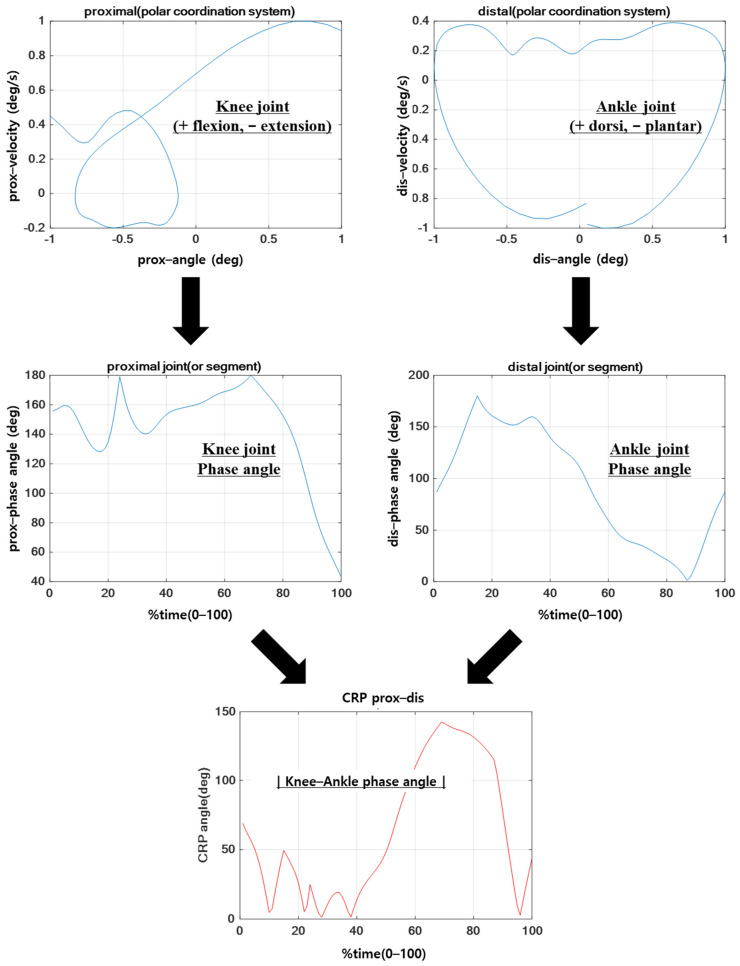
Procedure of continuous relative phase (CRP) data processing (the 1st row = each joint angular displacement and velocity on polar coordination system, the 2nd row = each joint phase angles, the 3rd row = CRP angle).

**Figure 4 sensors-21-00652-f004:**
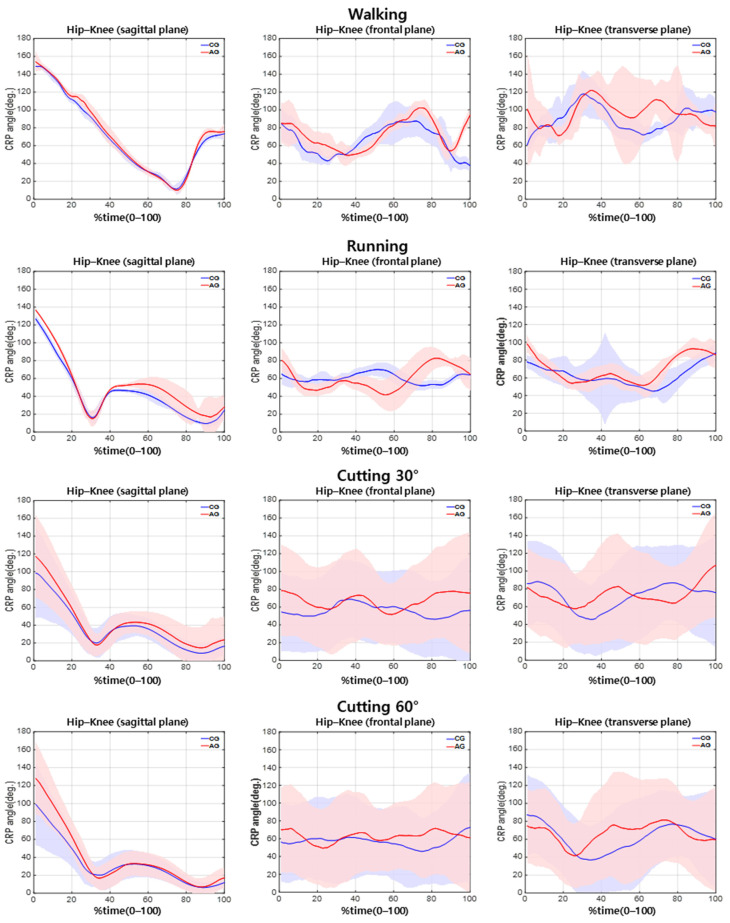
Ensemble average of CRP between hip–knee joint according to difficulty of movement.

**Figure 5 sensors-21-00652-f005:**
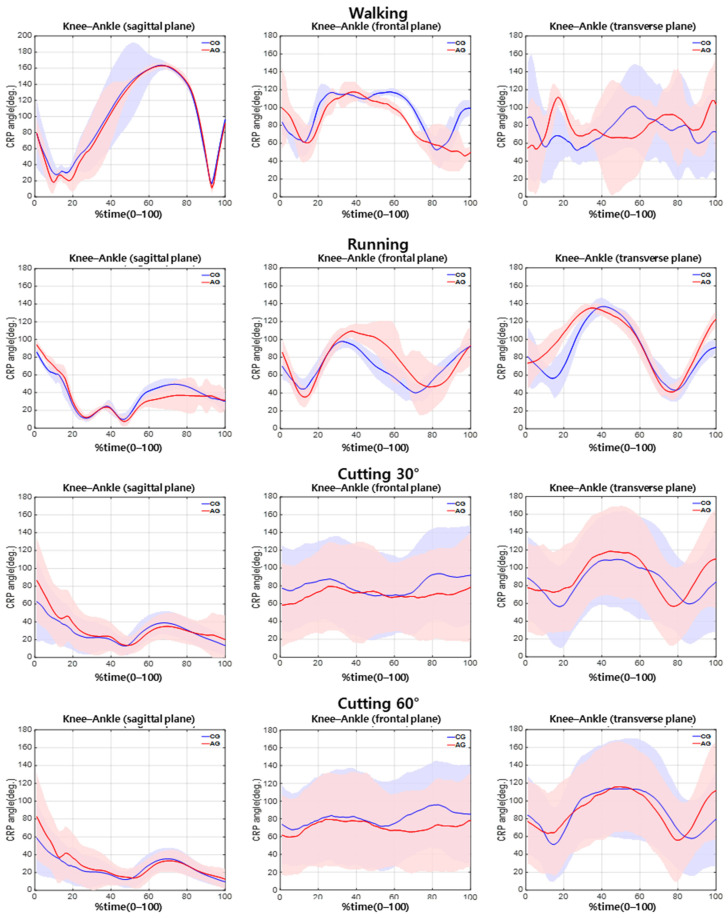
Ensemble average of CRP between knee–ankle joint according to difficulty of movement.

**Table 1 sensors-21-00652-t001:** Summary information of the participants.

Mean ± SD	Age (y)	Height (cm)	Weight (kg)	Period ACLR (mons.)	KOOS Score
CG (*n* = 21)	24.3 ± 1.7	175.0 ± 5.1	75.7 ± 7.8	-	-
AG (*n* = 21)	26.1 ± 4.0	177.6 ± 5.2	80.2 ± 7.1	49.1 ± 32.8	92.0 ± 4.9
t(*p*)	−1.86(0.07)	−1.61(0.12)	−1.95(0.06)		

**Table 2 sensors-21-00652-t002:** Hip–knee and knee–ankle CRP by movement difficulty.

Mean ± SD (Unit: deg)		CRP (Hip–Knee)			CRP (Knee–Ankle)		
		Sagittal Plane	Frontal Plane	Transverse Plane	Sagittal Plane	Frontal Plane	Transverse Plane
Walking	CG (*n* = 21)	70.03 ± 7.21	65.70 ± 29.73	90.57 ± 17.97	97.23 ± 11.29	93.78 ± 27.72	74.30 ± 24.65
	AG (*n* = 21)	72.61 ± 3.97	72.11 ± 25.14	96.30 ± 20.24	93.98 ± 15.53	83.60 ± 28.87	77.81 ± 23.52
	t(*p*)	−1.44(0.16)	−0.76(0.46)	−0.97(0.34)	0.78(0.44)	1.17(0.25)	−0.47(0.64)
Running	CG (*n* = 21)	42.87 ± 8.33	60.31 ± 28.15	62.33 ± 19.81	36.85 ± 7.31	67.79 ± 21.90	82.83 ± 22.80
	AG (*n* = 21)	50.41 ± 9.50	61.26 ± 27.64	69.64 ± 16.41	34.74 ± 7.48	74.64 ± 26.23	95.39 ± 16.68
	t(*p*)	−2.73(0.01) *	−0.11(0.91)	−1.30(0.20)	0.93(0.36)	−0.92(0.36)	−2.04(0.04) *
Cutting 30°	CG (*n* = 21)	35.94 ± 6.78	55.96 ± 18.57	72.07 ± 15.25	29.15 ± 6.32	80.43 ± 16.42	84.31 ± 14.44
	AG (*n* = 21)	41.87 ± 11.46	67.93 ± 20.06	72.91 ± 15.63	31.90 ± 7.44	70.46 ± 17.25	90.49 ± 15.00
	t(*p*)	−2.04(0.04) *	−2.01(0.05)	−0.18(0.86)	−1.29(0.20)	1.92(0.06)	−1.36(0.18)
Cutting 60°	CG (*n* = 21)	32.49 ± 5.54	56.68 ± 18.46	61.45 ± 13.92	25.60 ± 6.83	81.15 ± 14.72	87.21 ± 16.09
	AG (*n* = 21)	38.15 ± 7.35	62.56 ± 16.25	66.30 ± 13.80	28.66 ± 7.33	71.18 ± 14.48	88.57 ± 14.12
	t(*p*)	−2.82(0.01) *	−1.10(0.28)	−1.13(0.26)	−1.39(0.16)	2.21(0.03) *	−0.29(0.77)

* indicates significant difference; CRP, continuous relative phase.

**Table 3 sensors-21-00652-t003:** Hip–knee and knee–ankle CRP variability by movement difficulty.

Mean ± SD (Unit: deg)		CRP Variability (Hip–Knee)			CRP Variability (Knee–Ankle)		
		Sagittal Plane	Frontal Plane	Transverse Plane	Sagittal Plane	Frontal Plane	Transverse Plane
Walking	CG (*n* = 21)	6.10 ± 2.09	20.00 ± 8.08	23.85 ± 6.59	14.26 ± 4.77	19.93 ± 8.62	31.11 ± 9.74
	AG (*n* = 21)	6.31 ± 2.24	20.27 ± 7.09	23.93 ± 6.76	12.29 ± 3.16	21.84 ± 8.75	28.59 ± 6.88
	t(*p*)	−0.31(0.76)	−0.12(0.91)	−0.04(0.97)	1.57(0.12)	−0.71(0.48)	0.97(0.34)
Running	CG (*n* = 21)	5.70 ± 2.30	18.02 ± 7.54	20.61 ± 4.47	7.39 ± 2.43	17.79 ± 6.61	15.56 ± 3.83
	AG (*n* = 21)	6.28 ± 2.39	15.01 ± 4.15	22.11 ± 8.41	8.25 ± 2.29	16.28 ± 4.77	17.54 ± 7.80
	t(*p*)	−0.80(0.43)	1.60(0.12)	−0.72(0.48)	−1.18(0.24)	0.85(0.40)	−1.04(0.30)
Cutting 30°	CG (*n* = 21)	11.16 ± 3.43	32.92 ± 10.03	36.44 ± 9.59	12.05 ± 3.15	37.81 ± 8.06	39.93 ± 8.90
	AG (*n* = 21)	10.69 ± 3.27	35.62 ± 9.61	34.37 ± 6.57	11.80 ± 3.73	37.93 ± 7.26	35.20 ± 9.75
	t(*p*)	0.46(0.65)	−0.89(0.38)	0.82(0.42)	0.23(0.82)	−0.05(0.95)	1.64(0.10)
Cutting 60°	CG (*n* = 21)	9.85 ± 2.49	33.55 ± 9.16	31.29 ± 8.09	9.59 ± 2.77	40.34 ± 6.87	38.87 ± 10.01
	AG (*n* = 21)	7.88 ± 2.15	33.49 ± 8.61	33.91 ± 7.58	8.73 ± 2.35	38.80 ± 6.26	34.22 ± 9.60
	t(*p*)	2.74(0.01) *	0.02(0.98)	−1.08(0.29)	1.08(0.28)	0.75(0.45)	1.53(0.13)

* indicates significant difference; CRP, continuous relative phase.

## Data Availability

Not applicable.

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
