# Peer review of "Quantifying Coordination and Variability in the Lower Extremities after Anterior Cruciate Ligament Reconstruction"

_sensors, 2021, doi:10.3390/s21020652_

Round 1

Reviewer 1 Report

The authors provides an evualation method to determine the movement changes in the loweer limb coordination patterns according to move ment type following ACL  reconstruction by using a 3-dimensional motion analysis.  Results show that it can be useful to identify the problem and provide the fundamental evidence for the optimal timing of teturn-to-sport after ACLR rehabilitation. But there are still some comments as follows:

  1. Figure 1 can not explains the context from line 128 to line 131, that is, there is not any text or note to show the meanings in Figure1.
  2.  From line 164 to 181, the are some geomety parameters, such as angular displacement and angular velocity, that shall be explained in Figure1 or by other figures.
  3. The scope of this article is not suitable for publication in this journal because it is about the sports medicine, not the application of sensors or the topic of sensors.

Author Response

Dear Reviewer,

Thank you for your valuable comments and the opportunity to further revise the paper.

My co-author and I have thoroughly revised the manuscript to reflect the reviewer’s comments.

We are looking forward to your reply on revision. We are more than happy to make any further changes that will improve the paper. Thank you again for your attention and consideration.

The specific changes made in response to the reviewer's comments are detailed as the attached file.

Reviewer 2 Report

The project is interesting and could lead to a nice paper. Authors have a good methodology and probably interesting results. However, in the current form, they fail to present it in an intelligible way. Especially, the interest of the analysis of the coordination pattern is not enough developed while it should highlight motor control processes and potential issues with ACL. The method must be improved as well as the discussion.

Authors state that the literature is limited but they should search a second time, there is plenty article (and good articles) about coordination and relative phase in ACL or other troubles.

Authors can read the following paper to see how they could add values to their results. https://pubmed.ncbi.nlm.nih.gov/15756615/

The English must be edited by a native speaker scientist.

Introduction

  • The introduction brings too much different information without always having a real link between ideas. The introductory paragraphs are designed around information delivery, rather than issues. This leads to vague purpose of the paper. The knowledge gap needs to be made more explicit throughout the introduction. As it stands, it is difficult to appreciate the value of the study. The introduction needs major work to improve the structure of a progressive narrative towards this objective, by initially setting the broad context then refining towards a narrow gap in the field. For example, paragraph 1 should conclude with summary statement that reveals the broader problem, where paragraph 2 concludes with a statement that focuses on a narrow gap, while paragraph 3 can conclude by defining the explicit gap statement.

  • Several terms should be defined. It is strange to see a useless definition of what human locomotion is (page 2, line 55) when broad terms like “cutting maneuver” or ‘functional disorders” are not better defined.

  • Page 1, Line 43: “reduced static and dynamic stabilities, and reduced coordination”. Stability and coordination must be defined too.

  • Page 2, Line 47: “90% of muscle strength”. Which muscles?

  • Paragraph lines 47-53 does not bring real interesting information. Maybe authors should think how to formulate differently their purpose.

  • Page 2, Line 60: control group formed by healthy peers?

  • Paragraph lines 54-72 is all about results from previous articles but there is nothing to demonstrate how this information can be useful to explain and support the current work. Just looks like a list.

  • Page 2, Line 75: “one-dimensional aspects”. Do the authors mean analysis based on a unique feature or in a single plan of the movement?

  • Paragraph lines 73-80: Under this form, it is difficult to make a clear link between the statement on previous results (“however”) and the solution (“therefore”).

  • Sensors is a journal with broad topics. The description of the quantification of the CRP is not written in a way that non specialist can understand what it means really. Therefore, limited interest of your article, even for sport scientists or physiotherapists. I would recommend starting by phase portrait for one joint then relative phase.

  • Page 2, Lines 89-97: This paragraph is useless. It looks like it was written only because the journal is Sensors.

  • Overall, there is a real need to improve the rational about the coordination assessment and how it could bring valuable information. Hypotheses?

Methods

  • The criteria for inclusion are not enough restrictive for control subjects (“who had no history of lower limb injury within 6 months participated in this study”). It must be mentioned in the limitation. What about person with multiple strains for example?

  • Too many decimals in the table 1 leads to data unreadable.

  • Most important information is not presented. What was the resolution of the cameras (the reader does not have to know specification of all cameras on the market)? What are the dimensions of the useful capture area? For cutting maneuvers, how were represented the angle for the task to the participant?

  • Figure 1: add information about the view (back/front).

  • The explanation on page 5 is too limited in my opinion to let people not familiar with relative phase understanding the methods.

  • It is not fully clear what value of the CRP is taken. Is it the mean absolute value of the ensemble curve values calculated by averaging the absolute values of all points of the entire ensemble curve?

Discussion

  • The discussion is problematic: it repeats results without explaining really what it means, does not exploit nor interpret the coordination dynamics and uses a lot of shortcuts. I do not mean all justification or statement are wrong, however authors should be more careful and use more conditional structure of the verb. Paragraph lines 277-287 demonstrates a better structure.

  • A limitation paragraph is required.

Author Response

(The authors gave the same response as above.)

Round 2

Reviewer 1 Report

None

Reviewer 2 Report

Authors have made efforts to improve the overall manuscrit which now can be accepted under this form.